# Quantum Interference and Nonequilibrium Josephson Currents in Molecular Andreev Interferometers

**DOI:** 10.3390/nano10061033

**Published:** 2020-05-28

**Authors:** Noel L. Plaszkó, Peter Rakyta, József Cserti, Andor Kormányos, Colin J. Lambert

**Affiliations:** 1Department of Physics of Complex Systems, Eötvös Loránd University, Budapest 1095, Pázmány P. s. 1/A, Hungary; plaszkonoel@gmail.com (N.L.P.); rakytap@caesar.elte.hu (P.R.); cserti@elte.hu (J.C.); 2Department of Physics, Lancaster University, Lancaster LA1 4YB, UK

**Keywords:** superconductivity, molecular electronics, quantum interference, Cooper pair splitting

## Abstract

We study the quantum interference (QI) effects in three-terminal Andreev interferometers based on polyaromatic hydrocarbons (PAHs) under non-equilibrium conditions. The Andreev interferometer consists of a PAH coupled to two superconducting and one normal conducting terminals. We calculate the current measured in the normal lead as well as the current between the superconducting terminals under non-equilibrium conditions. We show that both the QI arising in the PAH cores and the bias voltage applied to a normal contact have a fundamental effect on the charge distribution associated with the Andreev Bound States (ABSs). QI can lead to a peculiar dependence of the normal current on the superconducting phase difference that was not observed in earlier studies of mesoscopic Andreev interferometers. We explain our results by an induced asymmetry in the spatial distribution of the electron- and hole-like quasiparticles. The non-equilibrium charge occupation induced in the central PAH core can result in a π transition in the current-phase relation of the supercurrent for large enough applied bias voltage on the normal lead. The asymmetry in the spatial distribution of the electron- and hole-like quasiparticles might be used to split Cooper pairs and hence to produce entangled electrons in four terminal setups.

## 1. Introduction

Quantum interference (QI) is ubiquitous in nature. Constructive quantum interference (CQI) leads to the formation of energy levels in atoms or molecules and energy bands in crystals, whereas destructive quantum interference (DQI) leads to energy gaps in molecules and band gaps in solids. The energy scale for these QI phenomena can be up to a few eV and therefore these quantum effects control the properties of molecules and solids at room temperature, for which $k_{B}T\approx 25$ meV ≪1 eV. In addition to these high temperature manifestations of QI, many low-temperature interference phenomena are well known, such as superfluidity and superconductivity, which occur on energy scale of order a few meV or less.

Investigations of QI in condensed systems are often driven by the desire to harness QI and deliver useful function. For example, when a molecule is placed into the nanogap between two metallic electrodes, it is known that electron transport from the source to the drain electrode is phase coherent at room temperature, provided the length of the molecule is less than approximately 3 nm. Consequently, if the interference pattern created by electronic de Broglie waves passing through the molecule can be controlled, then useful room-temperature devices such as molecular-scale switches, transistors and sensors could be realised. Single-molecule electronics is the sub-field of nanoelectronics [1,2,3,4,5,6,7], which aims to deliver such structures and in pursuing this goal, many groups have demonstrated that electrons can be injected into (and collected from) the core of a molecule with atomic accuracy [8,9,10,11]. Furthermore, it has been demonstrated that an ability to vary the atomic-scale connectivity to molecular cores is an effective way of controlling room-temperature QI [12,13]. On the other hand, at lower temperatures, quantum engineers strive to utilise QI in superconducting structures such as superconductor quantum interference devices (SQUIDs) and Andreev interferometers, which rely on controlling the interplay between a superconducting condensate and charge-carrying quasi-particles [14,15,16,17,18,19,20,21,22]. In such devices, QI is controlled by the phase of the superconducting order parameter, which describes a macroscopic collective degree of freedom, which has no counterpart at room temperature.

In this article, we examine the interplay between the high-energy-scale QI found in molecules and the low-temperature QI present in superconductors. Our aim is to determine how an ability to control the connectivity to molecular cores with atomic accuracy can be used to engineer the properties of Andreev interferometers and SQUIDS.

From the viewpoint of connectivity, a fundamental manifestation of QI is illustrated in Figure 1 top and middle panels, which shows an anthanthrene molecular core (consisting of 6 six-membered rings) connected by triple bonds to external electrodes. The connectivity of the triple bonds to the core is fixed by chemical synthesis. Figure 1 shows two examples of molecules with different connectivities. Following the numbering scheme of the lattice shown at the bottom of Figure 1, molecule 1 has triple bonds connected to atoms 12 and 3, whereas molecule 2 has triple bonds connected to atoms 9 and 22. The triple bonds are connected to terminal aryl rings, which in turn are connected to thioacetate anchor groups, which bind the molecules to source and drain electrodes. Since the triple bonds form weak links to the central core, it is conceptually convenient to consider the combination of an aryl ring, anchor group and external electrode as a single “compound electrode”, (coloured blue in Figure 1) which injects or collects electrons to or from the central core, via the triple bonds. Remarkably, when the external electrodes are normal (i.e., not superconducting), the room-temperature electrical conductance of setup 1 is both measured and predicted [23] to be approximately 81 times larger than that of setup 2. As explained in Reference [23], this conductance ratio is a clear manifestation of room-temperature QI. From the viewpoint of superconductivity, our aim is to replace one or more of the normal electrodes by superconducting electrodes and examine how electron transport though such molecular cores is controlled by a combination of connectivity and by the phase of the superconducting order parameter.

In ballistic normal-superconductor (*N*-*S*) hybrid systems the fundamental transport process is Andreev reflection, whereby an incoming electron is reflected back as a hole at the *N*-*S* interface. A rich set of physical phenomena that follow from this scattering process was realized in Andreev interferometers, which are devices with two (or more) superconducting and one (or more) normal leads attached to a central region [14,15]. For example, due to the extraordinary sensitivity of the Andreev current to the superconducting phase difference, Andreev interferometers may provide a faster and more precise alternative to superconductor quantum interference devices (SQUIDs) [16] to measure properties of quantum systems or even detecting Majorana bound states [17]. Importantly, the presence of a normal lead allows one to change the equilibrium occupation of Andreev bound states formed in multi-terminal *N*-*S* systems. It was suggested that such a non-equilibrium effect can be used to engineer π-Josephson junctions [18,19], where the fundamental relation $I_{s}=I_{c}\sin\left(\delta \Phi \right)$ between the phase difference δΦ of the order parameters of two superconductors and the supercurrent $I_{s}$ can be changed to $I_{s}=I_{c}\sin\left(\delta \Phi +\pi \right)$ ($I_{c}$ is the critical current). This effect has indeed been measured in diffusive meso-scopic multi-terminal *N*-*S* systems [20,21,22].

Recently, the superconducting properties of molecular-scale junctions have also started to attract experimental [24,25,26,27] as well as theoretical [28,29,30] interest. In Reference [30] we discussed the equilibrium properties of various multiterminal *N*-*S* systems where, in particular, QI effects in the core of the molecule play an important role. Here we show how such QI effects and non-equilibrium charge injection can lead to interesting effects in molecular Andreev interferometers. Namely, the non-equilibrium occupation of the Andreev bound states (ABSs) which are formed in superconductor-molecule-superconductor (*S*-*M*-$S^{\prime }$) Josephson junctions can be driven via the third, normal lead attached to the Josephson junction, thus realizing a non-equilibrium *N*-*M*-$SS^{\prime }$ system. As already mentioned, one of the key ingredients in our work is QI which arises in the molecular core of *N*-*M*-$SS^{\prime }$ systems that are based on polyaromatic hydrocarbons (PAHs) [12,13,23]. We find that in these systems one may observe effects that were not attainable is previously studied mesoscopic Andreev interferometers. Based on the “magic number theory of connectivity in References [12,23,30] we show that conductive channels through the molecular core can give rise to interfering paths contributing to the total ABS wave function with the same or with an opposite sign for electron and hole-like degrees of freedom. This rich set of interfering paths is provided by the conductive channels opened by the insertion of a substituent heteroatom into the molecular core [13]. Under specific circumstances the interplay of the interfering amplitudes may even lead to the total suppression of the electron-like (or hole-like) degrees of freedom on certain molecular sites and, at the same time, to a constructive interference for the hole-like (or electron-like) charge carriers. Since the charge current through the normal lead is closely tied to the Andreev reflection process, its magnitude is highly influenced by the density of both the electron- and hole-like particles in the vicinity of the normal contact. Thus, by measuring the charge current through the normal lead one can also probe the electron-hole separation in the molecular junction.

In what follows, we first describe the main characteristics of interference effects in Andreev interferometers based on PAHs [12,13,23] in equilibrium conditions. Our choice is justified by the peculiar mid-gap transport properties of these molecules accompanied by inner quantum interference effects within the core of the molecule [12,13,30,31,32,33,34,35,36,37,38,39,40,41,42]. We outline an illustrative connectivity-based theory that can be used to understand the current-phase relations at non-equilibrium conditions as well. Then we present our numerical results obtained for the normal and for the supercurrent at finite bias applied on the normal lead. We interpret our results in terms of connectivity arguments. We examine how the electrical properties of the Andreev interferometers would be influenced by tuning the inner QI effects of the molecular core. We demonstrate how QI can lead to a suppression of the normal current which is a clear evidence of the spatial separation of the electron- and hole-like particles. Finally, we present a summary of our most important results and give a brief outlook.

## 2. Results

From a conceptual viewpoint the key ingredients of our theoretical model are based on weak coupling, connectivity-driven, mid-gap transport and phase coherence. A detailed explanation of these assumptions is given in References [12,23,30]. Here we only mention that the term “weak coupling” means that the central aromatic molecule is weakly coupled to the contacts resulting in a small level broadening and self energy correction to the highest occupied molecular orbital (HOMO) and lowest unoccupied molecular orbital (LUMO) levels compared to the HOMO-LUMO gap. Thus, provided the Fermi level lies within the gap (resulting in near mid-gap transport), the quantum interference effects in the phase coherent transport processes are characterized by the properties of the molecular core alone. (The Coulomb interactions can be included at the level of a self-consistent mean field description such as Hartree, or Hartree-Fock.) Taken together, these conditions ensure that when computing the Green’s function of the core, the contribution of the electrodes can be ignored. Consequently, the probability of the propagation of the charged particles between sites *k* and *l* of the molecule is described by the “magic number” matrix element [12,23,30] $M_{kl}∼g_{kl}$, where $g\left(E_{F}\right)={\left(E_{F}-H_{\mathrm{mol}}\right)}^{-1}$ is the Green’s function of the isolated molecule described by Hamiltonian $H_{\mathrm{mol}}$. In particular, the electrical conductance corresponding to connectivity k,l is proportional to $|M_{kl}{|}^{2}$. In the simplest theoretical description an integer valued connectivity matrix $H_{\mathrm{mol}}$ captures the complexity of the inner CQI and DQI effects within the core of the molecule and when $E_{F}$ coincides with the middle of the HOMO-LUMO gap, the resulting magic number matrix $M_{kl}$ is simply a table of integers. In particular, for the bipartite lattice of Figure 1, when $E_{F}$ coincides with the middle of the HOMO-LUMO gap, destructive quantum interference arises between sites *k* and *l* that have the same parity (i.e., both are odd or both are even) and therefore the matrix element $M_{kl}$ is zero. In contrast, when the sites *k* and *l* have different parity, $M_{kl}$ may be finite giving a non-zero propagation amplitude of the charged particles between sites *k* and *l*. For the anthanthrene core of Figure 1, $M_{3,12}=9$ and $M_{9,22}=1$. Hence their conductance ratio is predicted to be $|M_{3,12}{|}^{2}/{\left|M_{9,22}\right|}^{2}=81$, which is close to the measured value of the conductance ratio, both for single molecules and for self-assembled monolayers [12,23,43].

### 2.1. Theoretical Background: Equilibrium Molecular Andreev Interferometers

In order to understand the unconventional non-equilibrium Andreev interference effect described in the next sections, following Reference [30] we first give a brief overview of the considerations that explain the interference pattern in the current $I_{N}$ flowing through the normal lead as a function of the superconducting phase difference $\delta \Phi =\phi_{1}-\phi_{2}$ between the S1 and S2 leads (see Figure 1) in equilibrium. Under equilibrium conditions, the limit eV→0 is understood, where *V* is the applied bias on lead *N* with respect to the chemical potential of the superconductors. Note that the imposition of a phase difference δΦ also generates a Josephson current $I_{s}$ flowing between the superconducting leads. Experimentally, as shown in References [44,45], δΦ can be controlled, thus allowing the measurement of the current-phase relation (CPR) of the supercurrent $I_{s}$ and the phase dependence of $I_{N}$.

Let us investigate $I_{N}$ in a device consisting of an anthanthrene central molecular core, as shown in Figure 1. The δΦ dependence of $I_{N}$ can be understood as a result of an interference effect between the possible transport paths of electrons and holes.

The arms of the interferometer are formed by the trajectories N→mol→S1→mol→N and N→mol→S2→mol→N. Let us consider the setup in which the normal lead *N* is attached to site labeled by 22, and the superconducting leads S1 and S2 are attached to sites 9 and 15, respectively and examine the transmission amplitude $t_{9,22}$ related to the process N→mol→S1→mol→N. Since the normal reflection on the superconductors does not give a contribution to the charge current, only Andreev reflection [46] can cause a net charge current. During Andreev reflection, an incoming electron-like quasiparticle is converted into a hole-like quasiparticle and vice versa at the normal-superconductor interface. Due to the Andreev reflection, an extra $e^{-i\phi_{1}}$ phase factor multiplies the transmission amplitude ($\phi_{1}$ is the phase of the superconductor S1). Then, the reflected hole-like state propagates back to the normal lead, a process which can be described by $-M_{9,22}$ according to the Bogolioubov-de Gennes equation [30]. Based on these considerations, the transmission amplitude can be written in the following form
(1)$$t_{9,22}∼-M_{9,22}^{2}e^{-i\phi_{1}}.$$

Similar considerations can be made for the transmission amplitude $t_{15,22}$. Since there are two interfering arms in the interferometer, one needs to sum up both transmission amplitudes associated with the two propagation paths to calculate the total transmission amplitude:(2)$$t_{tot}∼t_{9,22}+t_{15,22}=-M_{9,22}^{2}e^{-i\phi_{1}}-M_{15,22}^{2}e^{-i\phi_{2}},$$

From this expression the Andreev current $I_{N}$ at small bias voltage (eV≪Δ) can be approximated as:(3)$$I_{N}∼{\left|t_{tot}\right|}^{2}=M_{9,22}^{4}+M_{15,22}^{4}+2\cdot M_{9,22}^{2}\cdot M_{15,22}^{2}\cdot \cos\left(\phi_{1}-\phi_{2}\right).$$

As one can see, the Andreev current $I_{N}$ is indeed expected to show a simple dependence on the superconducting phase difference $\delta \Phi =\phi_{1}-\phi_{2}$ with a minimum at π. Regarding the supercurrent $I_{s}$ flowing between S1 and S2, in first approximation, this can be understood as a consequence of Andreev bound states (ABS), although a continuum of unbound states can also add a finite contribution [47].

### 2.2. Non-Equilibrium Numerical Calculations

To avoid time-dependent order parameter phases varying at the Josephson frequency, we assume that the superconductors S1 and S2 share a common condensate chemical potential μ. A finite bias voltage *V* (with respect to μ) can be then applied to the normal lead. This bias voltage will affect both the normal current $I_{N}$ and, by changing the equilibrium occupation of the ABSs, the supercurrent $I_{s}$ as well.

In order to describe the transport properties at finite bias voltage one has to use a theoretical framework capable of describing non-equilibrium transport processes. We calculate the currents $I_{N}$ and $I_{s}=\left(I_{S_{1}}-I_{S_{2}}\right)/2$ by using a tight binding approach and the Keldysh non-equilibrium Green’s function techniques [48,49,50]:(4)IN(Si)=−2ehRe∫dETrτ3ΓN(Si)G<(E),
with $\Gamma_{N(S_{i})}$ being the coupling from the molecule to the normal (superconducting) lead labeled by *N* ($S_{i}$) including the electron-hole degrees of freedom and $\tau_{3}$ is the third Pauli matrix acting on the electron-hole space. The current $I_{N(S_{i})}$ describes the current flowing through lead *N* ($S_{i}$) into the central molecule. In the steady state limit the currents flowing through the individual leads satisfy the charge conservation rule $I_{N}+I_{S_{1}}+I_{S_{2}}=0$ leading to two independent currents $I_{N}$ and $I_{s}$ characterizing the electrical properties of the junction. Finally, the lesser Green’s function $G^{<}$ in Equation (Equation 4) can be calculated within the Keldysh non-equilibrium framework using the Keldysh equation (see details in the Appendix C). The calculations were performed using the tight-binding framework implemented in the EQuUs [51] package. The relevant electronic states in the molecular core were described by a single orbital tight-binding model where the nearest neighbor sites are connected by a hopping amplitude $\gamma_{0}$. As shown in Figure 1, the hopping amplitude between the molecule and the normal *N* (superconducting S1, S2) lead is given by $W_{N}$ ($W_{S1}$, $W_{S2}$). In our calculations, unless indicated otherwise, we used $W_{N}=0.1\gamma_{0}$ and $W_{S}=0.3\gamma_{0}$. The normal and superconducting contacts were modeled by a one-dimensional tight-binding chain. The magnitude of the superconducting order parameter in the leads S1 and S2 was $\Delta =3\times 10^{-3}\gamma_{0}$. The results that we are going to discuss do not depend on the actual value of $\gamma_{0}$ and Δ. This simple model is justified by previous studies of connectivity driven transport processes through PAH molecules [12,23,30]. Following these works, our aim is to highlight the role of connectivity in the transport properties of these molecular cores leading to new interference phenomena. We give the remaining details of the tight binding-model used in our calculations in Appendix B.

### 2.3. Non-Equilibrium Molecular Andreev Interferometers

As a first example of non-equilibrium effects in Andreev-interferometers it is instructive to consider the system shown in Figure 2. With respect to Figure 1, we changed the connecting sites of the leads in order to “disarm” one of the interfering arms. This can be achieved by choosing connecting sites such that the magic number matrix elements between the sites connected to the normal lead and to one of the superconducting leads becomes zero, as shown in Figure 2. Therefore one may expect $I_{N}$ to be independent of the superconducting phase. Note, however, that the magic number $M_{6,9}$ between the superconducting leads is finite. Therefore, as we will show later, an ABSs is formed in this system and it has an important effect on $I_{N}$. The results for δΦ and eV dependence of $I_{N}$ can be seen in Figure 3a. The current $I_{N}$ remains very small for applied voltages eV≪Δ on the normal lead. As eV is increased, a finite $I_{N}$ starts to flow, but in contrast to the ∼cosδΦ dependence given in Equation (Equation 3), $I_{N}$ exhibits a *maximum* at superconducting phase difference δΦ=π.

These results can be explained by the effect of an ABS. As pointed out in earlier studies on multiterminal normal-superconductor mesoscopic systems [18,19], the voltage eV sets the effective electrochemical potential for the ABSs and those with energy $E_{n,ABS}\left(\delta \Phi \right)\leq eV$ are filled. The ABS energy $E_{n,ABS}\left(\delta \Phi \right)$ depends on the phase difference δΦ. A change in the occupation of the ABSs directly affects $I_{S1}$ and $I_{S2}$ and therefore the current distribution in the Andreev-interferometer junction will depend on both the voltage eV and on the phase difference δΦ. To illustrate these effects we show the supercurrent $I_{s}=\left(I_{S1}-I_{S2}\right)/2$ in Figure 3b. As eV is increased, a deviation from the simple $I_{s}=I_{c}\sin\left(\delta \Phi \right)$ relation can be clearly seen and for eV>Δ a π-transition takes place in $I_{s}$, similarly to what was obtained in References [18,19].

One can give a heuristic argument of why the presence of ABSs can affect $I_{N}$. This argument draws on analogies with the discussion given for the equilibrium case, that is, it is based on interfering quasiparticle trajectories. Although in the system depicted in Figure 2 the connectivity between the normal lead and the superconducting lead S2 is zero, the charge carriers can still probe the phase of lead S2 when they propagate along a path that also includes an Andreev reflection from the lead S1. Namely, as illustrated in Figure 2, both $M_{22,9}$ and $M_{6,9}$ are finite. We denote the amplitude of such propagation by $t_{6,22}^{\left(9\right)}$, where the upper index (9) indicates that the propagation between the sites 6 and 22 takes place via site the 9. To approximate the amplitude $t_{6,22}^{\left(9\right)}$ one can make similar considerations as in the previous section. Thus,
(5)$$t_{6,22}^{\left(9\right)}∼\left(-M_{22,9}\right)\cdot e^{-i\phi_{1}}\cdot M_{9,6}\cdot e^{i\phi_{2}}\cdot \left(-M_{6,9}\right)\cdot e^{-i\phi_{1}}\cdot M_{9,22}.$$

This amplitude describes a (a) propagation from the normal lead to the superconducting electrode S1 ($M_{9,22}$), (b) Andreev reflection from electrode S1 ($e^{-i\phi_{1}}$), (c) propagation of the hole-like state from contact S1 to S2 ($-M_{6,9}$), (d) Andreev reflection of the hole-like particle on the contact S2 ($e^{i\phi_{2}}$), (e) electron-like propagation between the superconducting electrodes S1 and S2 ($M_{9,6}$), (f) a third Andreev reflection on the superconducting electrode S1 ($e^{-i\phi_{1}}$), (g) and a hole-like propagation from the contact S1 to the normal lead ($-M_{22,9}$). Finally, we also take into account in our minimal model the amplitude $t_{9,22}$ describing the direct propagation between the normal lead and the lead S1 according to Equation (Equation 1). The observed interference effect can be explained as the interplay between these two amplitudes:(6)$$I_{N}∼{\left|t_{9,22}+t_{6,22}^{\left(9\right)}\right|}^{2}=M_{9,22}^{4}\cdot \left(1+M_{9,6}^{4}-2\cdot M_{9,6}^{2}\cdot \cos\left(\phi_{1}-\phi_{2}\right)\right).$$

The normal current $I_{N}$ in Equation (Equation 6) has a maximum at phase difference $\phi_{1}-\phi_{2}=\pi$. The minus sign appearing in front of the $\cos(\phi_{1}-\phi_{2})$ term in Equation (Equation 6) is due to the peculiar properties of the Bogolioubov-de Gennes quasiparticles. Namely, the amplitude $t_{6,22}^{\left(9\right)}$ contains one more hole-like propagation compared to the amplitude $t_{9,22}$, which brings in an extra minus sign needed for the formation of the unconventional interference effect. Note, that this argument does not explain why the increase in $I_{N}$ appears only above a certain bias voltage. Moreover, the transport process associated to the amplitude $t_{6,22}^{\left(9\right)}$ contains four more tunnelings between the superconducting leads and the molecular core compared to the amplitude $t_{9,22}$. Thus, the amplitude $t_{9,22}$ might be expected to be much larger than the amplitude $t_{6,22}^{\left(9\right)}$ which would suppress the interference effect between these two interfering paths.

The role of the ABSs can be shown explicitly by using Green’s function theory to calculate the differential conductance $\frac{dI_{N}}{deV}$. The details of this calculation are presented in the Appendix C. For simplicity, let us assume that there is only one ABS in the system. (The general case of more than one ABS is also discussed in the Appendix C). Then the differential conductance can be approximated as [52]
(7)$$\frac{dI_{N}}{deV}\approx \frac{16e}{h}\frac{\Gamma_{ABS}^{e}\Gamma_{ABS}^{h}}{{\left(eV-E_{ABS}\right)}^{2}+\Gamma_{ABS}^{2}},$$
where $\Gamma_{ABS}=\Gamma_{ABS}^{e}+\Gamma_{ABS}^{h}$ is the level broadening of the ABS due to the presence of the normal lead, ΓABSe=〈ABS,e|WN†Im(gNe)WN|ABS,e〉 with $W_{N}$ being the coupling between the normal lead and the central molecule, $W_{N}^{†}$ is the Hermitian conjugate of $W_{N}$, $g_{N}^{e}$ stands for the electron-like block of the surface Green’s function of the normal contact evaluated at energy $E_{ABS}$, Im(⋯) denotes the imaginary part of a function, and |ABS,e〉 represents the electron-like components of the wave function of the ABS. The definition of $\Gamma_{ABS}^{h}$ is analogous to $\Gamma_{ABS}^{e}$ involving the hole-like degrees of freedom instead of the electron-like components. According to Equation (Equation 7), the ABS leads to a resonant peak of Lorentzian lineshape in the differential conductance for $eV\approx E_{ABS}\left(\delta \Phi \right)$. The half-width of the resonance is determined by the finite lifetime of the ABS which is due to the coupling to the normal lead given by $\Gamma_{ABS}^{e}$ and $\Gamma_{ABS}^{h}$. We note that a similar result can be obtained for a system hosting multiple ABSs. The total differential conductance in this case would be a sum of resonances centered on the energies of the individual ABSs’. However, the “cross-talk” between the ABSs has an additional influence on the shape of the resonances, that is, they start to deviate from the regular Lorentzian shape. (For details see the Appendix C.)

Looking back to Equation (Equation 6), one may now say that the interfering amplitude $t_{6,22}^{\left(9\right)}$ can be increased due to the Fabry-Perot-like resonant oscillations of the charged particles between the superconducting contacts. These oscillations lead to the formation of ABSs of finite lifetime, which, in turn, affect the current $I_{N}$ at finite eV, as indicated by Equation (Equation 7).

The ABSs can be visualized by calculating the density of states of the junction (see Appendix D for details). The results of such calculations for the system in Figure 2 are shown in Figure 4. In Figure 4a,b we show the density of states for two different coupling $W_{N}$. The large values of the density of states (bright region) indicate the ABS. In this particular case, for eV=0 and zero temperature there is only one occupied ABS (at energy −E, not shown) and one unoccupied ABS [at energy *E*, Figure 4a,b]. By applying a finite eV>0 the occupation of these ABSs can be changed, leading to the peculiar dependence of both $I_{N}$ and $I_{s}$ on δΦ in Figure 3 that we noted earlier. Because of the normal lead, the ABSs have a finite lifetime, which is determined by the escape rate of the particles through the normal lead. Therefore, the ABS lifetime (and consequently the width of the resonant peaks in the differential conductance) is expected to be sensitive to the coupling between the normal lead and the central molecule. This can be clearly seen in Figure 4c,d, where the peak of the differential conductance calculated for $W_{N}=0.1\gamma_{0}$ is considerably narrower than the peak calculated for $W_{N}=0.3\gamma_{0}$. Notice, that for higher values of $W_{N}$ the resonant peak starts to deviate from the Lorentzian shape. This is because by increasing the coupling between the contacts and the central molecule one can no longer neglect the energy dependence of the Green’s function of the normal contact in the calculation of $\Gamma_{ABS}^{e}$ and $\Gamma_{ABS}^{h}$, see Appendix C for details. Since the ABSs energy $E_{ABS}$ depends on the phase difference δΦ, the peaks in $\frac{dI_{N}}{deV}$ are also sensitive to the superconducting phase difference. This is also shown in Figure 4c,d. Therefore, by measuring $\frac{dI_{N}}{deV}$ as a function of δΦ one may obtain spectroscopic information about the ABSs [18].

The role of ABSs and QI in the molecular core can be further illustrated by studying the finite bias properties of the system shown already in Figure 1, bottom panel. For this configuration of the leads the magic number vanishes between the two sites where the superconducting electrodes are attached. One may therefore expect that there is no ABS present in the system. According to our calculations this is not exactly the case: one can find an ABS whose energy is very close to the value of the pair potential Δ in the leads, but it is nearly independent of δΦ and therefore it can carry only a small supercurrent. This explains that for a finite bias eV the δΦ dependence of $I_{N}$ remains qualitatively the same as in the zero bias case discussed in Equations (Equation 1)–(Equation 3) and shows a minimum at δΦ=π for all bias voltages (Figure 5a).

The supercurrent $I_{s}$ shows the conventional ∝sinδΦ dependence for eV<Δ [Figure 5b]. By comparing Figure 5a,b, one can see that although a small $I_{s}$ can flow for finite eV, the critical current $I_{c}$ is smaller than $I_{N}$. This is the opposite of what we found in the previous case [Figure 3]. Overall, one may also notice that both $I_{N}$ and $I_{c}$ are much smaller than previously, c.f. Figure 3 and Figure 5.

These results underpin the importance of ABSs in Andreev interferometers and are also a consequence of mid-gap transport. Namely, the propagation amplitude described by the Green’s function elements decay with the energy difference between the chemical potential and the energy of the eigenstates. Since the energy levels of the molecule are much further from the chemical potential than the ABS levels, their contribution to the Green’s function elements will be also much smaller than the contribution of the ABSs. Thus, in the mid-gap energy regime, the transport processes would indeed be dominated by the interference effects related to the ABSs.

We now discuss the most general situation, where ABSs can be found in the system and, in contrast with the case in Figure 1, the connectivity from the normal lead to both superconducting terminal is finite. In what follows we shall examine the unconventional interference effects as the connectivity in the interferometer is changed. We consider a setup similar to the one shown in Figure 1 and tune the asymmetry of the molecular interferometer by inserting a substitutional heteroatom into the molecular core [13], that is, a carbon atom is replaced by a substituent heteroatom, as indicated schematically in Figure 6a. Due to the presence of the heteroatom, new conductive channels open up in the molecular core that were originally closed via destructive QI effects. In our theoretical model we account for the presence of a substitutional heteroatom by a modified on-site energy on a specific site in the molecule. By changing for example, the on-site energy $\varepsilon_{3}$ in the tight-binding Hamiltonian of the molecular core [see Figure 6a], the normal conductance between sites labeled by even numbers also becomes finite [13]. Assuming that instead of S1 and S2 we have normal conducting leads N1 and N2 as in Figure 6a, the evolution of the ratio of the zero-bias normal conductances $\sigma_{N,N1}$ and $\sigma_{N,N2}$ as a function of the on-site energy $\varepsilon_{3}$ is demonstrated in Figure 6b. As one can see, by varying $\varepsilon_{3}$ one can gradually open a conductive channel between leads *N* and N2.

We now consider the finite bias properties of the Andreev interferometer shown in Figure 6c, which can be obtained be replacing the normal leads N1 and N2 by superconducting ones S1 and S2 in Figure 6a. First, we calculate $I_{N}$ for several values of $\varepsilon_{3}$ and fixed eV=0.95Δ. Remarkably, as shown in Figure 7a, $I_{N}$ takes on a hat-like shape with two maxima around the phase difference π for such values of $\varepsilon_{3}$, where $\sigma_{N,N1}$ and $\sigma_{N,N2}$ are comparable. This is clearly different from the results in Figure 3a and we are not aware of similar results in mesoscopic NS systems. Regarding $I_{s}$ [Figure 7b], for smaller values of $\varepsilon_{3}$ where $\sigma_{N,N1}\gg \sigma_{N,N2}$, it is qualitatively similar to the results shown in Figure 3b. On the other hand, the current-phase relation of $I_{s}$ becomes similar to the conventional $I_{s}=I_{c}\sin\delta \Phi$ as the conductive channel gradually opens between *N* and S2 and consequently $\sigma_{N,N1}\approx \sigma_{N,N2}$ [see for example, the case $\varepsilon_{3}=-0.50\gamma_{0}$ in Figure 7b]. Note that in this case the $I_{s}\left(\delta \Phi \right)$ dependence for δΦ≈π is different from the corresponding eV=0.94Δ result shown in Figure 3b.

In Figure 7c,d we show $I_{N}$ and $I_{s}$, respectively, as a function of δΦ for several biases eV. Here we fixed $\varepsilon_{3}=-0.50\gamma_{0}$, that is, $\sigma_{N,N1}\approx \sigma_{N,N2}$. As one can see, for small eV, when the occupation of the ABS is not yet modified, $I_{N}$ shows qualitatively the same behavior as in Figure 5a, that is, when there was no current-carrying ABS in the system. For larger eV, however, there is a clear difference with respect to both Figure 3a and Figure 5a, since $I_{N}$ adopts a hat-like dependence on δΦ. The non-equilibrium population of the ABSs also affects $I_{s}$ [see Figure 7d] which starts to deviate from the ∝sinδΦ dependence for eV>0.95Δ.

According to our calculations the presence of a heteroatom does not modify the ABS spectrum significantly [Figure 8a]. As shown in Figure 8b, when $I_{N}$ nearly vanishes for $\varepsilon_{3}=-0.50\gamma_{0}$, δΦ=π [Figure 7a], the differential conductance $\frac{dI_{N}}{deV}$ vanishes, too. According to Equation (Equation 7), the vanishing of $\frac{dI_{N}}{deV}$ can be explained only if the coupling between the normal lead and the ABS vanishes.

Therefore we turn our attention to the electron- and hole-like broadening terms $\Gamma_{n}^{e}$ and $\Gamma_{n}^{h}$ in the numerator of Equation (Equation 7). In Figure 8c,d we show the local density of states (LDOS) on the molecular site connected to the normal contact separately for the electron- and hole-like degrees of freedom. Note that $\Gamma_{n}^{e}$ and $\Gamma_{n}^{h}$ are proportional to the corresponding LDOS. As one can see, the electron-like component of the LDOS becomes highly suppressed at phase difference δΦ=π, while the hole-like components has a maximum there. In turn, we found that on other sites of the molecule the hole-like component of the LDOS can be suppressed and the electron-like LDOS enhanced (an example is shown in Appendix E).

The surprising result in Figure 8c,d can be understood as a peculiar interference effect that acts in a different way on the electron- and the hole-like particles. (Note, that due to electron-hole symmetry, the same feature can be observed for negative energies with a constructive interference in the electron-like part of the LDOS and with a destructive interference in the hole-like part of the LDOS.) One can give the following simple argument in terms of new quasiparticle paths. In Figure 9 we show two quasiparticle trajectories. The first trajectory, shown in Figure 9a, describes the process N→S2→S1→N and the last segment S1→N is made possible by the fact that the substitutional heteroatom opened a new conductive channel. Using a similar argument as in the case of Equation (Equation 5), one can argue that the amplitude of the path contributing to the electron-like part of the wave function can be expressed as
(8)$$\begin{array}{ccc} t_{\mathrm{odd}}^{e} & ∼ & M_{N,S1}\cdot e^{i\phi_{1}}\cdot \left(-M_{S1,S2}\right)\cdot e^{-i\phi_{2}}\cdot M_{S2,N}\\ & ∼ & -M_{N,S1}\cdot M_{S1,S2}\cdot M_{S2,N}e^{i(\phi_{1}-\phi_{2})}. \end{array}$$

Since $t_{\mathrm{odd}}^{e}$ contains an odd number of propagations through the molecule, and the sign of the propagation depends on whether one considers electron- or hole-like particles, the amplitude $t_{\mathrm{odd}}^{h}$ contributing to the hole-like component of the wave function would differ by a minus sign compared to $t_{\mathrm{odd}}^{e}$. Now consider the process N→S2→S1→S2⇒N depicted in Figure 9b. The last propagation indicated by S2⇒N describes a normal reflection (without electron-hole conversion) at the site connected to S2 and a forthcoming propagation to the normal contact. (Since we are working in the weak coupling limit, the normal reflection at sites connected to the contacts has a finite probability.) The amplitude corresponding to this path can be expressed as follows:(9)$$\begin{array}{ccc} t_{\mathrm{even}}^{e} & ∼ & M_{N,S2}\cdot M_{S2,S1}\cdot e^{i\phi_{1}}\cdot \left(-M_{S1,S2}\right)\cdot e^{-i\phi_{2}}\cdot M_{S2,N}\\ & ∼ & -M_{N,S2}^{2}\cdot M_{S1,S2}^{2}\cdot e^{i(\phi_{1}-\phi_{2})}. \end{array}$$

Since $t_{\mathrm{even}}^{e}$ depends on the square of the connectivity matrix elements, the corresponding hole-like amplitude $t_{\mathrm{even}}^{h}$ would have the same sign as $t_{\mathrm{even}}^{e}$. One can see that because of the sign difference, there is a destructive interference in total amplitude $t^{e}=t_{\mathrm{even}}^{e}+t_{\mathrm{odd}}^{e}$ and a constructive one in $t^{h}=t_{\mathrm{even}}^{h}+t_{\mathrm{odd}}^{h}$. This example shows how differences can appear in the processes that determine the electron-like and the hole-like LDOS. Note that, strictly speaking, in the calculation of $t_{}^{e}$ and $t^{h}$ one would need to take into account all possible scattering paths and not only those discussed above. We expect, however, that the described interference effect would not be affected significantly.

We also mention that according to our numerical results the interference effect can be swapped between the electron- and hole-like components by changing the sign of the on-site energy $\varepsilon_{3}$ of the heteroatom. According Equation (Equation 8) of Reference [13] the connectivity matrix element $M_{N,S1}$ can change a sign for sufficiently large heteroatom on-site energy. Consequently, $t_{odd}^{e}$ would also change sign resulting in a constructive interference for the electron-like and destructive interference for the hole-like components in the LDOS.

Opening of new conductance channels can affect the properties of the molecular Andreev interferometer not only in the case discussed in Figure 6 and Figure 7, where the conductance between the leads *N* and S1 was tuned. As noted earlier for the system in Figure 1, for this configuration of the leads the connectivity matrix element is zero between the two sites where the superconducting electrodes are attached. However, this connectivity matrix element can also be made finite by adding a heteroatom as indicated in Figure 10a. This means changing the onsite energy $\varepsilon_{12}$ in the tight-binding Hamiltonian of the molecular core. We found that the dependence of the supercurrent on $\varepsilon_{12}$ and eV is qualitatively similar to the behavior in Figure 7b,c. Therefore we only show the calculations for $I_{N}$ in Figure 10b. As the connectivity grows for larger values of $\varepsilon_{12}$, the δΦ dependence of $I_{N}$ also undergoes a drastic change and, interestingly, adopts a qualitatively similar behavior to the one shown in Figure 7a, that is, there are two maxima in the current around δΦ=π.

## 3. Conclusions

In this article, we have investigated the interplay between two quantum interference phenomena that take place on hugely different energy scales; QI within molecules, which takes place on the scale of electron volts and QI associated with superconductivity, which takes place on the scale of milli-electron volts. We studied the interplay between connectivity-driven QI in molecular cores and non-equilibrium charge distribution in three-terminal Andreev interferometers based on PAH molecules. We showed that QI determines certain fundamental properties of the ABS in the system, while their energies can be tuned by the phase difference between the superconducting probes. Consequently, QI and the non-equilibrium ABS occupation in the molecular core, which can be modified by a bias voltage applied to the normal lead, affects both the normal current and the supercurrent in the system. We gave a simplified explanation of some of the complicated interference effects in terms of electron and hole trajectories and point out when such explanation breaks down under non-equilibrium conditions. We found that the dependence of the normal current on the superconducting phase difference can exhibit a double-peak structure, while the supercurrent can show a π transition when the bias eV on the normal lead is larger than the superconducting gap. We also showed that adding a heteroatom to the PAH core can significantly change the QI and can induce an asymmetry in the spatial distribution of the electron- and hole-like particles, which has a direct impact on the phase dependence of the normal current. This indicates that the properties of molecular Andreev interferometers can be tuned by engineering QI in the molecular core.

For the future one may envisage a system similar to the one shown in Figure 6 but with two normal leads (N1 and N2) attached to different sites of the molecular core. Assume now that lead N1 would be coupled to a site where, for example, the electron LDOS is enhanced and the hole suppressed, whereas lead N2 to a site where the opposite is true, that is, the electron LDOS is suppressed and the hole LDOS is enhanced. Then the so-called non-local Andreev reflection (N1→N2), where an incoming electron from lead N1 is Andreev reflected into lead N2, would be enhanced with respect to local Andreev reflection (N1→N1) and normal electron transmission (N1→N2). Therefore, in such four-terminal device the asymmetry between the electron- and hole-like degrees of freedom on certain sites of the molecular core could be translated into a spatial separation of electron pairs originating from the superconducting condensate. This process is called Cooper pair splitting and it provides entangled electron pairs that may play an important role in quantum information processing. Most of the proposed Cooper pair splitters to-date relied on Coulomb blockade transport through quantum dots [53,54,55,56], or on peculiar properties of novel low-dimensional materials [57,58,59]. Our results indicate that Cooper pair splitting may also be achieved in multi-terminal molecular systems where the spatial separation of the Cooper pairs would rely on the inner QI effects of the molecule. The detailed study of such four-terminal molecular Cooper pair splitters is an interesting problem which we leave to a future work.

## Figures and Tables

**Figure 1 nanomaterials-10-01033-f001:**
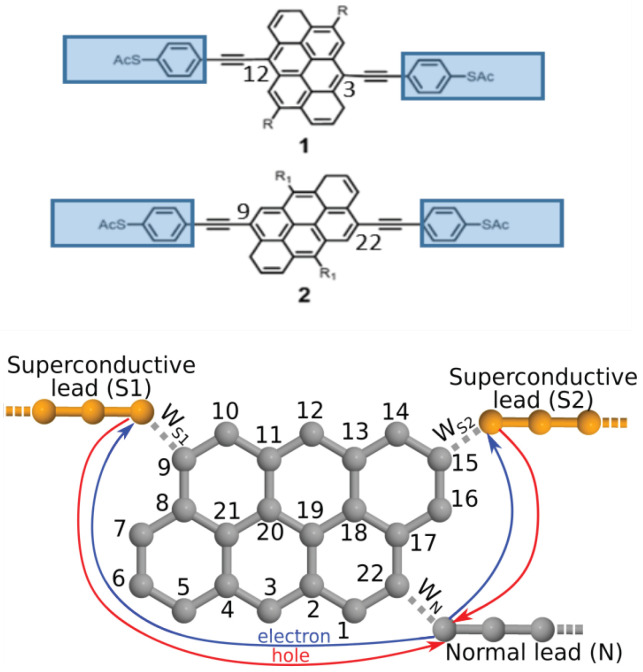
The top and middle panels show molecules 1 and 2, with connectivities 12,3 and 9,22 to the anthanthrene molecular core. The lower panel shows an Andreev interferometer consisting of an anthanthrene molecule, two superconducting leads and one normal lead. The “sites” of the associated tight-binding model represent p${}_{z}$ orbitals of the anthanthrene molecule and are labelled according to the figure. The coupling of the molecule to the normal (superconducting) lead is denoted by $W_{N}$ ($W_{S}$), for details see the text. The superconducting leads are characterized by a superconducting order parameter $\Delta e^{i\phi_{1}}$ and $\Delta e^{i\phi_{2}}$. The transport processes responsible for the conventional interference effect are indicated by solid lines for the electron-like (blue) and hole-like (red) propagation.

**Figure 2 nanomaterials-10-01033-f002:**
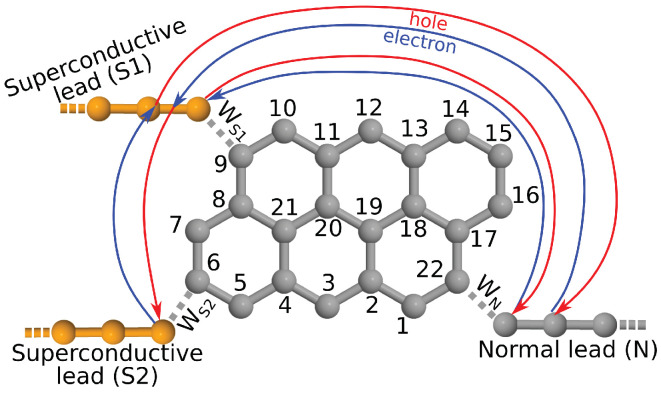
Anthanthrene molecule attached to two superconductive and one normal lead. The connectivity matrix element between the sites 6 and 22 is zero, while the connectivity between sites 9 and 22 and between sites 9 and 6 is finite. Solid lines indicate the propagation of the electron-like (blue) and hole-like (red) quasiparticles.

**Figure 3 nanomaterials-10-01033-f003:**
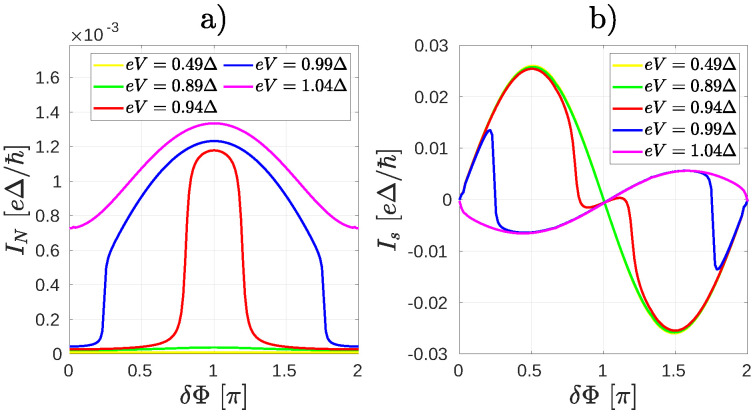
The currents $I_{N}$ (**a**) and $I_{s}$ (**b**) as a function of the phase difference δΦ between the superconducting leads for the system depicted in Figure 2 for several bias voltage eV. (**a**) a robust peak in $I_{N}$ appears around the phase the difference π when the bias voltage is comparable to the superconducting gap Δ. (**b**) the supercurrent shows a π transition for eV>Δ.

**Figure 4 nanomaterials-10-01033-f004:**
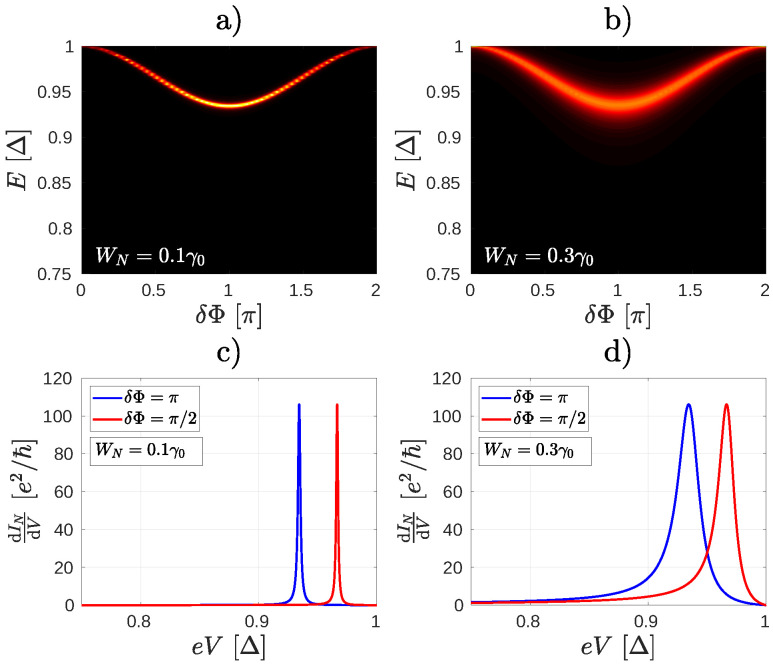
(**a**,**b**) Density of states of the molecular junction shown in Figure 2 for two different coupling $W_{N}$ of the normal lead to the molecule. The bright areas indicate the dispersion of the Andreev Bound State (ABS) as a function of δΦ. The ABS energy level is broadened by increasing $W_{N}$. (**c**) A resonance occurs in the differential conductance when the bias voltage eV is close to the energy $E_{n}\left(\delta \Phi \right)$ of the ABS in (**a**). (**d**) As the ABS is broadened, the width of the resonance broadens as well.

**Figure 5 nanomaterials-10-01033-f005:**
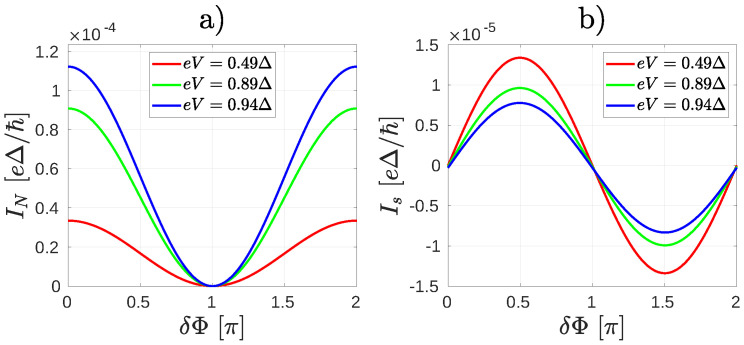
The currents $I_{N}$ (**a**) and $I_{s}$ (**b**) as a function of the phase difference δΦ between the superconducting leads for the system depicted in Figure 1 for several bias voltage eV. (**a**) the Andreev current $I_{N}$ shows a minimum at δΦ=π. (**b**) The current-phase dependence of supercurrent is $I_{s}\propto \sin\delta \Phi$.

**Figure 6 nanomaterials-10-01033-f006:**
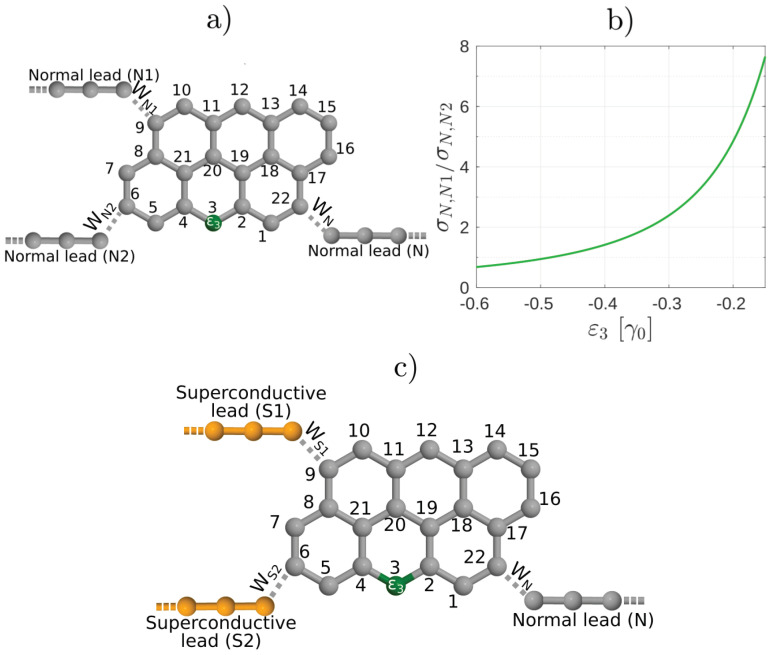
(**a**) Schematics of Anthanthrene molecule with a heteroatom denoted by green. (**b**) Ratio of the normal conductance between contacts N−N2 and N−N1 as a function of the on-site energy $\varepsilon_{3}$ in Figure 6. At $\varepsilon_{3}=0$ the conductance $\sigma_{N,N1}$ between contacts *N* and N1 is much larger than the conductance $\sigma_{N,N2}$ between contacts *N* and N2, in agreement with References [12,23]. For finite $\varepsilon_{3}$ the conductance $\sigma_{N,N2}$ increases and can be comparable to $\sigma_{N,N1}$. (**c**) Andreev interferometer setup obtained by replacing the normal leads N1 and N2 in (**a**) by superconducting ones S1 and S2.

**Figure 7 nanomaterials-10-01033-f007:**
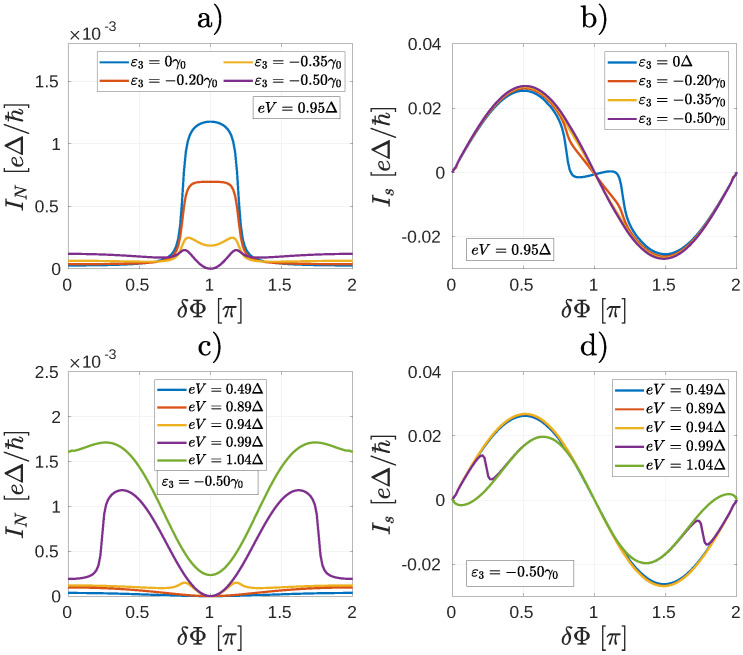
The currents (**a**) $I_{N}$ and (**b**) $I_{s}$ (**b**) as a function of the phase difference δΦ between the superconducting leads for the system depicted in Figure 6a for several values of the on-site energy $\varepsilon_{3}$ and fixed bias eV=0.95Δ. $I_{N}$ starts to show a double peak structure as a function of δΦ for $\varepsilon_{3}\approx -0.20\gamma_{0}$. (**c**) $I_{N}$ and (**d**) $I_{s}$ as a function of the bias voltage eV for $\varepsilon_{3}=-0.50\gamma_{0}$.

**Figure 8 nanomaterials-10-01033-f008:**
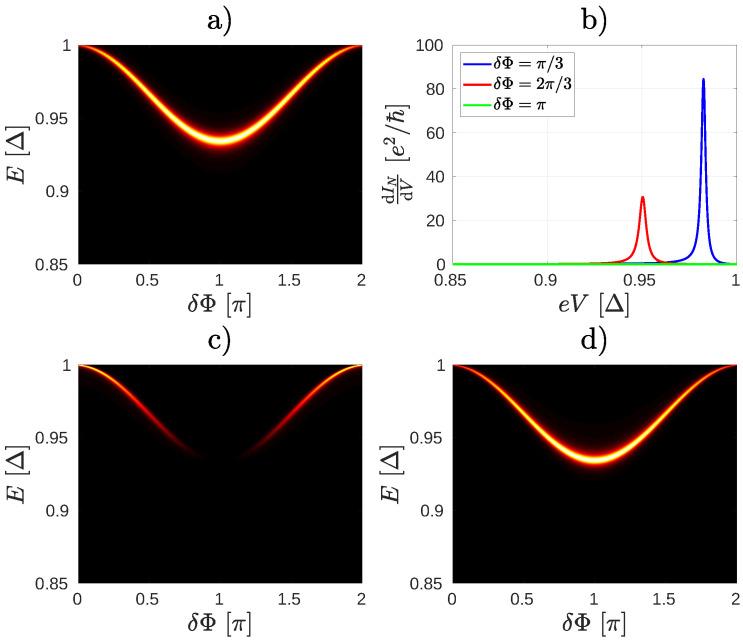
(**a**) The density of states of the ABS for the case shown in Figure 6c. (**b**) The differential conductance corresponding to (**a**). The local density of states for electron (**c**) and hole (**d**) quasiparticles as a function of δΦ. In these calculations we used $\varepsilon_{3}=-0.50\gamma_{0}$.

**Figure 9 nanomaterials-10-01033-f009:**
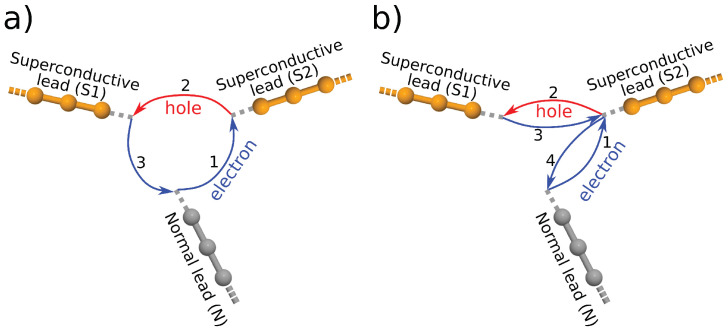
(**a**) An example for interfering paths having an amplitude of opposite sign for the electron- and hole-like particles. These kinds of paths have an odd number of propagation through the molecular core. (**b**) A representative of trajectories consisting of even number of propagations through the molecular core. The amplitude of these kinds of trajectories have the same sign for the electron- and hole-like quasiparticles.

**Figure 10 nanomaterials-10-01033-f010:**
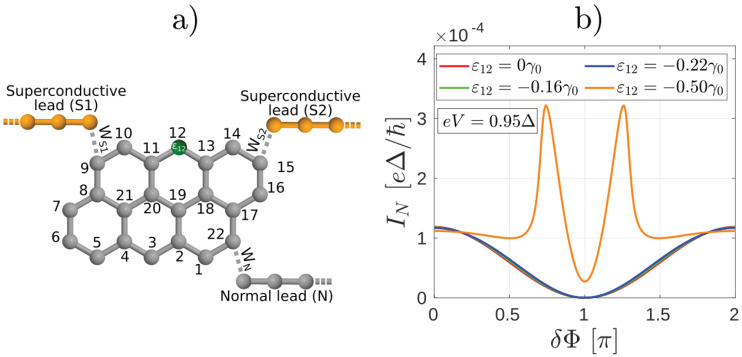
(**a**) Anthanthrene molecule with a heteroatom (denoted by green) and the same configuration of leads as in Figure 1. (**b**) The normal current $I_{N}$ as a function of the phase difference δΦ between the superconducting leads for the setup in (**a**).

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
