# Peer review of "Quantum Interference and Nonequilibrium Josephson Currents in Molecular Andreev Interferometers"

_nanomaterials, 2020, doi:10.3390/nano10061033_

Round 1

Reviewer 1 Report

The authors report on a theoretical study of the current in a three terminal device consisting of a molecular core, two superconducting contacts and one normal one. Using tight binding model and the Keldysh non-equilibrium Green's function technique, the authors calculate the normal and superconducting current independently. They show that electrical contact position and the incorporation of a heteroatom can greatly modify the normal and superconducting currents in the device.

I think the authors have presented a clear and intriguing investigation of this molecular system. The combination of molecular-based quantum interface combined with superconducting quantum interference effects is interesting and the proposed Cooper pair splitter is indeed a fun idea. I would recommend publication of this article provided the authors consider the following minor details.

- On page 2, lines 55-58. It was not readily apparent to me that quantum interference is why the conductance in setup 1 would be higher than setup 2. If I simply look at the models, I am led to believe that the current path through 1 is shorter than 2 and therefore 1 would have a higher conductance. I would suggest entering Ref 30. into the discussion here as support for these introductory statements.

- On page 4, lines 133-134, if I have understood Ref. 30, I think the matrices should be M3,12 = 9 and M9,22 = 1. Also, the ratio should be adjusted in the same line.

- On page 8, lines 280-281. "This can be clearly seen in Fig. 4c) and d), where the peak of the differential conductance calculated for WN = 0.1g0 is considerably wider than the peak calculated for WN = 0.3g0." It's the other way around, 0.3 is wider.

Author Response

Reply to Reviewer 1

We are happy to hear that the Reviewer recommends the publication of our manuscript. We thank the Reviewer for the careful reading of the manuscript and bringing to our attention the minor points listed in her/his report. We answer her/his remarks point-by-point.

1) "On page 2, lines 55-58. It was not readily apparent to me that quantum interference is why the conductance in setup 1 would be higher than setup 2. ... I would suggest entering Ref 30. into the discussion here as support for these introductory statements."

We agree with the Reviewer that the sentences in lines 55-58 should be re-formulated so that the role of quantum interference is more apparent. We have re-written the corresponding sentences and cite Ref.30.

2) " On page 4, lines 133-134, if I have understood Ref. 30, I think the matrices should be M3,12 = 9 and M9,22 = 1. Also, the ratio should be adjusted in the same line."

We thank the referee for spotting this error in the manuscript. The correct values are indeed M3,12 = 9 and M9,22 = 1, we have corrected it.

3) "This can be clearly seen in Fig. 4c) and d), where the peak of the
differential conductance calculated for WN = 0.1g0 is considerably wider than the peak calculated for WN = 0.3g0." It's the other way around, 0.3 is wider."

This was indeed an error, we have corrected it.

Author Response

Reply to Reviewer 2

We are happy to hear that the Reviewer recommends the publication of our manuscript. We thank the Reviewer for the careful reading of the manuscript and bringing the minor points listed in his report to our attention. We answer her/his remarks point-by-point.

1) "The phase difference \delta \phi defined in lower-case in page 4 but later in page 8 or in the Figures it is presented in the upper-case."

We now follow the notation used in the figures, i.e., \delta\Phi, throughout in the text.

2) "The values of the superconducting order parameter \Delta are inconsistent in the main text, in page 5 and in the Appendix in page 15."

We have used \Delta=3 10^{-3}\gamma_0 in our calculations. The corresponding value in the Appendix A has been corrected.

3) " The notation of the level broadenings are not consistent, ..."

We have made sure that the notations mentioned by the Reviewer are now consistent.

4) "In page 8, 4th line from the bottom, it said "the peak of the differential conductance calculated for W_N=0.1\gamma_0 is considerably wider than the peak calculated  for W_N=0.3\gamma_0". "

This was indeed an error, we have corrected it.

5) "In Fig.A1, I cannot understand the phrase "The sites in the molecular core are labeled by primed and unprimed numbers". What are primed numbers? "

This sentence referred to a previous version of Fig.A1 and we forgot to update it.

We have now updated Appendix A and the caption of Fig.A1.

6) Small grammatical typos: we thank the referee for communicating them to us, we have corrected them.